

# New role for an old acquaintance: miR-1246 as a new inflammatory and prognostic marker in polytrauma patients

Liudmila P. Leppik, Melissa Manamayil, Cora Schindler, Ramona Sturm, Philipp Störmann, Dirk Henrich, Ingo Marzi and Birte Weber

Department of Trauma Surgery and Orthopedics, Johann Wolfgang Goethe Universität Frankfurt am Main, Frankfurt, Hessen, Germany

## ABSTRACT

**Background**. Based on a literature analysis, we hypothesized that miR-1246 has a high potential as new biomarker after trauma. This miRNA is already established in oncology but has not yet been described in polytrauma.

**Methods**. Plasma samples from polytraumatized patients with an ISS ≥ 16 were collected in the emergency room (ER) and 48 hours after trauma. The patients were divided into two groups: a group affected by polytrauma with a leading traumatic brain injury (TBI) (abbreviated injury scale head, AIShead > 4) and a group with a polytrauma without TBI (AIShead = 0). The expression of miR-1246 was measured using qRT-PCR in plasma and plasma extracellular vesicles (EVs). Lastly, we isolated CD171 + EVs by using a magnetic bead-based method and measured miR-1246 expression.

**Results**. In plasma, there was a significant increase in miR-1246 in the ER in polytrauma patients, but not in TBI patients. The EV miRNA expression was also significantly increased in the ER samples of the polytrauma patients ($*p \leq 0.0001$), while an increase in the expression in the TBI patients ($*p \leq 0.01$) was only observed after 48 hours. The systemic expression of miR-1246 correlated with the Injury Severity Score (ISS), creatine kinase and creatinine kinase MB (CK-MB), myoglobin, Interleukin (IL)-6 and the length of hospital stay. In CD171+neuro-EVs, the miR-1246 expression was also significantly increased.

**Conclusion**. MiR-1246 was shown to be a marker for the patients' injury severity, the early inflammatory phase and the patients' outcome.

Corresponding author
Birte Weber, bi.weber@med.uni-frankfurt.de

## INTRODUCTION

Polytrauma continues to have a high socioeconomic impact and mortality. For the clinical outcome of patients, it is crucial to quickly and precisely detect the severity of the injury, damaged organs and the occurrence of adverse events (*Hardy et al., 2023*; *Meakes et al., 2024*). Several biochemical markers have been employed in the clinical settings (*Frink et*

*al., 2009*; *Wang et al., 2018*; *Alexiou & Voulgaris, 2024*). Yet, the search for novel specific biomarkers is ongoing.

In the last years, blood miRNAs have emerged as potential biomarkers in multiple types of diseases because the expression patterns of blood miRNAs reflect tissue injuries, biochemical and physiological imbalances. MiRNAs are short, non-coding RNA molecules, which are characterized by a high stability, specificity and selectivity; and therefore, might qualify as a good alternative to protein biomarkers (*Etheridge et al., 2011*). Multiple studies have been conducted to identify specific miRNA profiles in critically ill polytrauma patients (*Bedreag et al., 2015*). The list of identified damage-specific miRNAs is permanently growing. As of now, these studies are scarce due to the high complexity of the clinical presentation of polytrauma patients and the challenges to model polytrauma *in vivo* and in vitro. In contrast, in the field of cancer research, multiple cellular and *in vivo* models have significantly speeded up the research process. These models have shed light on the important role of multiple miRNAs. miRNAs, are involved in major pathways during carcinogenesis, but it is also assumed that they affect the body's systemic immune and inflammatory responses in polytrauma patients. We performed a literature analysis to narrow down and identified a potential miRNA candidate: miR-1246. First identified in human embryonic stem cells (*Morin et al., 2008*), this miRNA is involved in carcinogenic events in multiple tissues and it sexpression is regulated by p53 (*Zhang et al., 2011*). Its oncogenic role was shown in colorectal, breast, renal, oral, laryngeal, pancreatic and ovarian cancers as well as melanoma and glioma. miR-1246 reportedly regulates the activity of RAF/MEK/ERK, GSK3β, Wnt/β-catenin, JAK/STAT, PI3K/AKT, THBS2/MMP and NOTCH2 pathways and stimulates proliferation, cell cycle progression, migration and invasiveness of cancer cells (reviewed in *Ghafouri-Fard et al., 2021*). Apart from the systemic appearance of miR-1246, very little is known about the miR-1246 in extracellular vesicles (EVs). In patients with amyotrophic lateral sclerosis miR-1246 was deregulated in EVs and was associated with disability progression (*Saucier et al., 2019*). In exosomes released by melanoma and neuroblastoma cells, miR-1246 was associated with the development of metastasis through the transfer of miR-1246 to lung macrophages, liver macrophages, and stellate cells. In this metastatic context, miR-1246 induced an inflammatory reaction in the early phase of metastatic progress due to the EV-transfer of miRNA (*Blavier et al., 2023*).

Very little is known about the role of this miRNA in other diseases and traumatic injuries than cancer. miR-1246 was suggested to mediate ALI-induced lung inflammation *via* nuclear factor 'kappa-light-chain-enhancer' of activated B-cells (NF-κB) activation *in vitro* and *in vivo* (*Suo et al., 2018*). The pro-angiogenic role of exosomal miR-1246 from endothelial progenitor cell was shown in a myocardial infarction model *in vivo* (*Huang et al., 2021*). In contrast, the anti-angiogenic role of exosomal miR-1246 from stem cells of human deciduous exfoliated teeth (SHED-Exos) was shown in HUVEC cells (*Liu et al., 2022*). In one study, miR-1246 was increased in patients with community-acquired pneumonia and pneumonia-related sepsis (*Hermann et al., 2020*). Another study with patients showed that miR-1246 is downregulated in sepsis and sepsis-acute lung injury. Lipopolysaccharide (LPS) treatment was found to decrease miR-1246 expression levels in

human bronchial epithelial cells (*Xu et al., 2021*). Taken together, the literature implicates a connection between inflammation and miR-1246.

Therefore, we hypothesized that miR-1246 expression is increased due to the body's inflammatory response in polytrauma patients. To validate this, we compared the expression of this miRNA in plasma and in plasma EVs among different groups of polytrauma patients and healthy controls.

## MATERIALS & METHODS

### Study design

The university local ethics committee granted ethical approval (approval ID 89/19 and 375/14) for all experiments, which were carried out in compliance with STROBE principles (*von et al., 2008*) and the Declaration of Helsinki. For enrolled patients and volunteers, formal written informed consent was acquired. The study includes traumatized patients admitted to the university hospital level 1 trauma center (Germany) between 2016 and 2020.

Table 1 lists the demographic and clinical features of the study's participants, which includes 10 patients with major traumatic brain injury (TBI, ISS $\geq$ 16, AIShead $\geq$ 4, $n = 10$), 10 polytrauma patients without TBI (PT, ISS $\geq$ 16, AIShead $= 0$, $n = 10$), and 10 healthy volunteers. Out of the polytrauma patients' cohort, two groups of patients, with low ISS (ISS $< 20$, $n = 12$) and high ISS (ISS $\geq 20$, $n = 11$) were sorted out. The ISS is the sum of the squared AIS scores from the three most severely injured body regions. Clinical data and outcome parameters were collected by using the digital patients record of the included patients.

After admission to the emergency room, blood samples were taken 24 and 48 h later, stored on ice, and centrifuged for 15 min at 3,500 rpm (4 °C) to extract plasma.

### EV and (CD171+) neuro -EV isolation

For EV isolation, plasma was cleared *via* a 30 min centrifugation at 16,000 g (4 °C). EVs were isolated from 100 µl of plasma by using size exclusion chromatography (SEC) (EX03-50, Cell guidance system, Cambridge, UK) and characterized as described previously (*Weber et al., 2023*; *Weber et al., 2024*).

Neuro-EV isolation was performed by means of Exosome-Streptavidin Isolation/DetectionReagent (Thermo Fisher Scientific, Waltham, MA USA) and biotinylated CD171 antibody (eBio5G3; eBioscience™, Waltham, MA USA). To do so, four µg biotinylated anti-CD171 antibody were coupled with one ml ($1 \times 10^7$) magnetic beads following the manufacturer's protocol. Afterwards the 100 µl SEC- isolated EVs were individually mixed with 100 µl coupled beads and incubated overnight at 4 °C and neuro-EV were collected according to the protocol (Fig. S1) and directly used for miRNA isolation.

### miRNAs isolation from plasma and EV

miRNAs were isolated with miRNeasy Serum/Plasma Advanced Kit (Qiagen Inc., Hilden, Germany) according to the manufacturers protocol from either 200 µl plasma, 180 µl EVs or 100 µl neuro-EVs. miRNAs were used for cDNA synthesis performed with

**Table 1  Mean clinical and laboratory parameters of the TBI and PT patients.**

| | TBI ($n = 10$) | PT ($n = 10$) | P value[#] |
|---|---|---|---|
| Gender (male in %) | 60 | 100 | – |
| ISS | 28.5 (3.93) | 35.27 (13.27) | ns |
| Plasma miR-1246, ER | 0.08 (0.15) | 0.98 (0.98) | ns |
| Plasma miR-1246, 48 h | 0.008 (0.009) | 0.071 (0.167) | ns |
| EV miR-1246, ER | 0.52 (1.349) | 0.588 (0.43) | $p = 0.011$[*] |
| EV miR-1246, 48 h | 0.32 (0.328) | 0.05 (0.06) | $p = 0.008$[**] |
| CK (U/l) | 476.1 (584.89) | 1265.8 (1315.3) | $p = 0.027$[*] |
| CK-MB (U/l) | 49.5 (41.52) | 101.7 (68.89) | $p = 0.035$[*] |
| Myoglobin (ng/ml) | 197.5 (197.96) | 2203.3 (1132.0) | $p = 0.0003$[***] |
| Troponin T (pg/ml) | n.a. | 64.2 (111.4) | – |
| Leukocytes (per µl) | 9330 (4720) | 11970 (3100) | ns |
| Creatinine (mg/dl) | 0.811 (0.265) | 1.089 (0.16) | $p = 0.035$[*] |
| IL-6, ER (pg/ml) | 65.08 (±137.78) | 183.42 (121.65) | $p = 0.0089$[**] |
| IL-6, 48 h (pg/ml) | 65.54 (63.45) | 533.9 (1011.43) | ns |
| IL-10 ER (pg/ml) | 39.49 (59.55) | 157.36 (171.26) | $p = 0.02$[*] |
| IL-10 48 h (pg/ml) | 13.65 (25.74) | 8.8 (±12.84) | ns |
| C-reactive protein (mg/dl) | 0.316 (0.393) | 0.14 (0.156) | ns |
| Hemoglobin (g/dl) | 12.8 (1.46) | 11.06 (2.35) | ns |
| Lactate (mg/dl) | 24.87 (11.41) | 31.75 (18.89) | ns |
| Survival (%) | 80 | 100 | – |
| Dismissal (%) | Home: $n = 1$ Rehabilitation clinic: $n = 6$ Death: $n = 2$ Hospice: $n = 1$ | Home: $n = 2$ Rehabilitation clinic: $n = 7$ Other hospital: $n = 1$ | – |
| Time in hospital (days) | 21 (12.97) | 37.2 (23.36) | ns |
| Need for catecholamines (%) | 30 | 54.54 | |
| Ventilation time (days) | 9.55 (12.4) | 3.375 (2.59) | ns |
| Time on ICU/IMC (days) | 13.37 (13.12) | 17.88 (13.71) | ns |

Notes.
[#]Statistical analysis *via* Mann–Whitney-Test.
[*]$p \leq 0.05$.
[**]$p \leq 0.01$.
[****]$p \leq 0.0001$.
ns, non significant.
miR-1246 expression presented as $2^{-dCT}$ values.

miRCURY LNA RT Kit (Qiagen Inc., Hilden, Germany) according to the manufacturer instructions with addition of cel-miR-39-3p as Spike-in control. RT-qPCR amplification was performed with the miRCURY SYBR Green PCR Kit (Qiagen Inc., Hilden, Germany) and commercially available primers (hsa-miR-1246 and cel-miR-39-3p; miRCURY LNA TM miRNA PCR Assay, Qiagen Inc., Hilden, Germany). Amplification was performed in the CFX96 Touch Real-Time PCR Detection System (Bio-Rad, Heidelberg, Germany) as follows: one cycle with 3 min at 95 °C, and 40 cycles with 10 s at 95 °C and 50 s at 56 °C. All reactions were performed in duplicates and the delta Ct method ($2-\Delta$Ct) was used for relative quantification of miR-1246 using the Ct values of spike-in cel-miR-39-3p.

### ELISA

IL-6 concentrations in plasma (ER, 24 h and 48 h time points) were measured by using Quantikine ELISAs (Human IL-6: # D6050; R&D Systems, Minneapolis, USA) according to the manufacturers' protocol. IL-10 concentrations (ER, 24 h and 48 h time points) were measured by mean of Quantikine ELISA (Human IL-10: #D1000B, R&D Systems, Minneapolis, USA) according to the manufacturer's protocol.

### Statistical analysis

All statistical analyses were conducted with Graph Pad-Prism 9 (Dotmatic, San Diego, CA, USA). The Kruskal–Wallis test and Dunn's multiple comparisons test were used to examine the data. The Mann–Whitney test was used to statistically analyze two groups. The Spearman rank correlation was used for correlation analysis. If $r > 0.5$, the correlation was considered moderate; if $r > 0.7$, it was considered high. Statistically significant was set to 5% ($p < 0.05$). For data visualization, the mean $\pm$ standard error of the mean (SEM) is displayed.

## RESULTS

### miR-1246 in polytrauma and TBI

First, we compared the expression of plasma and EV miR-1246 in healthy controls and two groups of trauma patients (polytrauma and TBI) at ER and 48 h post trauma (Fig. 1, Table 1). Our results show that at ER, but not at 48 h, the systemic level of miR-1246 was significantly increased in polytrauma patients compared to controls (Fig. 1A). EV miR-1246 was also significantly increased in polytrauma patients at ER compared to healthy controls. At 48 h, it was only significantly increased in TBI patients compared to healthy controls (Fig. 1B).

Comparative analysis of clinical and laboratory parameters of both groups of patients showed that polytrauma patients were characterized by significantly higher levels of CK, CK-MB, myoglobin, creatinine and IL-6 concentration at ER. At the same time, these patients had significantly lower level of IL-10 measured at ER (Table 1).

Next, we analyzed if the increase of miR-1246 expression reflected the severity of trauma and compared its systemic and EV expression among polytrauma patients with ISS > 20, with ISS < 20 and healthy controls. miR-1246 was expressed significantly higher in patients with more severe trauma at both time points (Fig. 2A) compared to healthy controls, while a significant increase of miR-1246 expression was also observed in patients with ISS < 20 at the ER compared to healthy controls. In case of EV miR-1246, the expression was significantly upregulated only in ISS > 20 patients' group at ER time point (Fig. 2B).

A comparison of the clinical parameters of these patient groups revealed significant differences in ISS, CK, CK-MB, myoglobin, IL-6 measurements at ER and 24 h. In addition, patients with ISS > 20 experienced a significantly longer time at ventilation (Table 2).

To verify if the miR-1246 expression indeed reflects the inflammation and injury severity in patients, we performed a correlation analysis of the miR-1246 expression and patients' clinical data and routine laboratory parameters. The results (Fig. 3) showed that miR-1246 plasma expression strongly correlates with ISS ($r = 0.75$, $p < 0.001$) when measured at

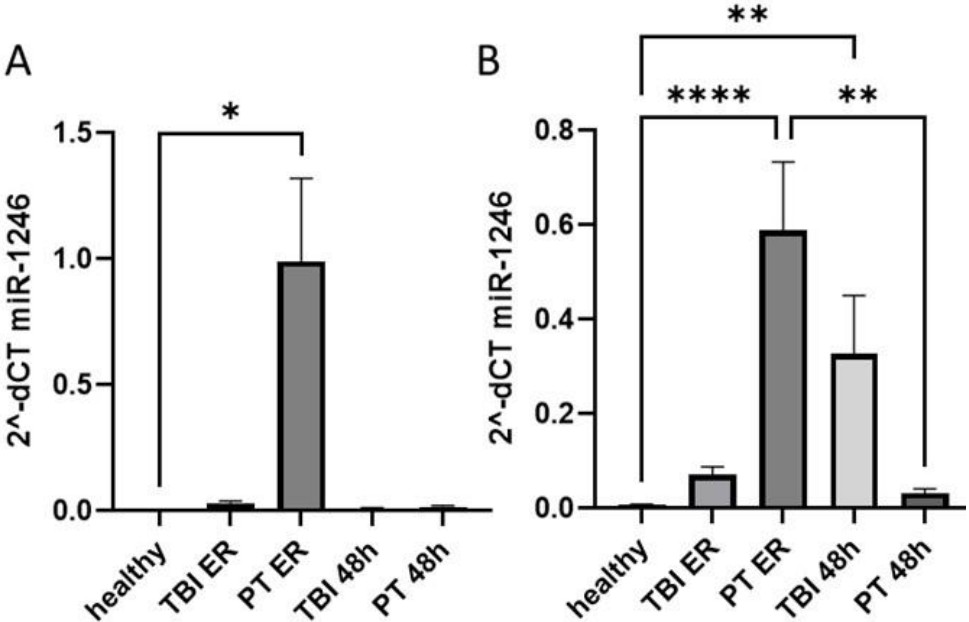

**Figure 1** **miR-1246 expression in plasma and EVs.** (A) miR-1246 expression measured *via* qRT-PCR in plasma of patients with leading traumatic brain injury (TBI) and polytraumatized patients (PT) compared with healthy controls in the emergency room (ER) and 48 h after trauma; (B) miR-1246 expression in extracellular vesicles (EVs). *$p \leq 0.05$, **$p \leq 0.01$, ****$p \leq 0.0001$.

ER and moderately correlated at 24 h time point ($r = 0.64$, $p < 0.01$). At ER, it is also moderately correlated with IL-6 ($r = 0.66$, $p < 0.001$), myoglobin ($r = 0.68$, $p < 0.001$), CK ($r = 0.43$ $p < 0.05$), CK-MB ($r = 0.49$, $p < 0.05$) and time in hospital ($r = 0.46$, $p < 0.05$). In the group of TBI patients, plasma miR-1246 strongly correlated with leukocytes ($r = 0.76$, $p < 0.05$) and EV miR-1246 had a strongly negative correlation with lactate ($r = -0.85$, $p < 0.05$).

### miR-1246 in CD171+neuro-EVs

To investigate the details of the miR-1246 dynamic in different compartments, CD171+ neuro-EVs were isolated by a magnetic-bead-based method. The expression of miR-1246 in these neuro-EVs was significantly increased in polytrauma patients in the ER (Fig. 4). The miR-1246 concentration in CD171+EVs is negatively correlated with the ISS ($r = -0.75$), lactate ($r = -0.7$), CK-MB ($r = -0.82$, $p < 0.05$). The concentration positively correlated with the time on the ICU/IMC ($r = 0.9$).

### DISCUSSION

miR-1246 is a well-characterized miRNA in the context of cancer, and is known to be implicated in inflammatory pathways, such as NF-kB signaling (*Jiang et al., 2022*). Based on the known role of miR-1246 in inflammation and the fact that polytraumatized patients suffer from significant immune dysfunction (*Li et al., 2023*), we hypothesized that systemic miR-1246 expression could be relevant in the context of trauma (*Pape et al., 2022*). To

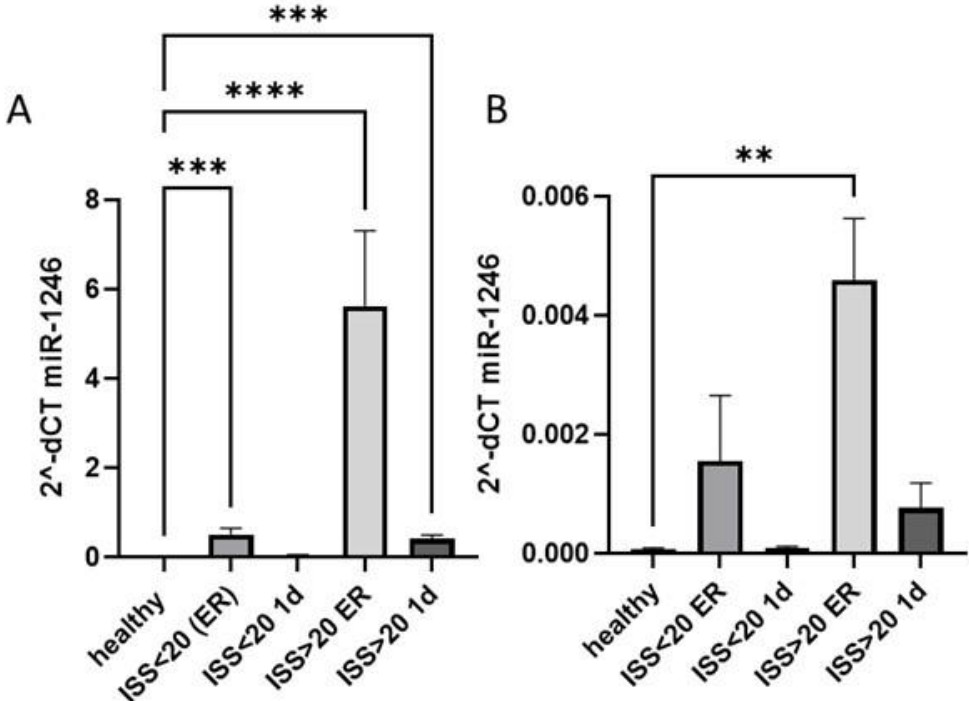

**Figure 2** **miR-1246 expression in plasma and EVs.** (A) miR-1246 expression measured *via* qRT-PCR in plasma of polytraumatized patients with an Injury Severity Score (ISS) < 20 and > 20 compared with healthy controls in the ER and 24 h after trauma; (B) miR-1246 expression in extracellular vesicles (EVs). $**p \leq 0.01$, $***p \leq 0.001$, $****p \leq 0.0001$.

validate this hypothesis, we compared the expression of miR-1246 in plasma, in systemic EVs and CD171+neuro-EVs in different groups of trauma patients and healthy controls.

## Polytrauma patients

First, we observed that the systemic level of miR-1246 was increased in both groups of patients, with isolated TBI and polytraumatized patients without TBI, at ER (significant only in polytrauma patients). The increase in miR-1246 compared to controls was much higher in polytrauma patients. Because of the more serious trauma severity in polytrauma patients, we hypothesized that the injury severity or greater extent of tissue damage explain the higher expression of miR-1246 (although ISS is not significantly different in PT *vs.* TBI). Therefore, we divided the patients in two subgroups according to the injury severity score (ISS > or <20) and compared the expression levels of miR-1246 in these groups. As expected, polytrauma patients with higher ISS experience a significantly higher miR-1246 expression; there is a significant correlation ($r = 0.75$) among the initial miR-1246 expression and the ISS. Further, we observed that this increase in polytraumatized patients correlated strongly with the early systemic inflammatory response detected by the systemic IL-6 concentration. In general, ISS is described as a very effective approach for evaluating prognosis, mortality and complications in trauma patients (*Yadollahi et al., 2020*). IL- 6 is also known as a strong predictor of patient's outcome (*Sapan et al., 2016*). A more severe injury could be

**Table 2 Mean clinical and laboratory parameters of polytraumatized patients with Injury Severity Score (ISS) <20 and >20.**

| | ISS <20 ($n = 12$) | ISS >20 ($n = 11$) | P value[#] |
|---|---|---|---|
| Gender (male in %) | 83.3 | 63.3 | – |
| ISS | 17.33 (0.47) | 42.1 (8.64) | $p < 0.0001$[****] |
| Plasma miR-1246, ER | 0.81 (1.08) | 14.9 (26.8) | $p = 0.001$[***] |
| Plasma miR-1246, 24 h | 0.17 (0.41) | 3.06 (7.56) | $p = 0.0005$[***] |
| CK (U/l) | 237.08 (137.18) | 557.2 (390.06) | $p = 0.0106$[*] |
| CK-MB (U/l) | 57.83 (21.73) | 161.7 (93.55) | $p = 0.0004$[***] |
| Myoglobin (ng/ml) | 331.50 (427.65) | 1558.7 (1444.92) | $p = 0.0007$[***] |
| Troponin T (pg/ml) | 67.05 (91.78) | 208.3 (388.46) | ns |
| Leukocytes (per μl) | 11 560 (3070) | 15 872 (6397) | ns |
| Creatinine (mg/dl) | 0.86 (±0.30) | 1.065 (0.38) | ns |
| IL-6, ER (pg/ml) | 57.01(39,57) | 213.51 (179.76) | $p = 0.0005$[***] |
| IL-6, 24 h (pg/ml) | 57.69 (39.9) | 190.7 (126.29) | $p = 0.012$[*] |
| IL-10, ER (pg/ml) | 67.71 (88.88) | 104.68 (87.34) | ns |
| IL-10, 24 h (pg/ml) | 5.02 (6.29) | 9.81 (6.17) | ns |
| C-reactive protein (mg/dl) | 0.27 (0.49) | 0.421 (0.38) | ns |
| Hemoglobin (g/dl) | 12.64 (1.66) | 13.27 (1.92) | ns |
| Lactate (mg/dl) | 18.33 (7.51) | 26.22 (14.83) | ns |
| Survival (%) | 100 | 100 | – |
| Dismissal | Home: $n = 10$<br>Rehabilitation clinic: $n = 2$ | Home: $n = 4$<br>Rehabilitation clinic: $n = 6$<br>Other department: $n = 1$ | – |
| Time in hospital (days) | 13.92 (7.02) | 22.6 (19.53) | ns |
| Need for catecholamines (%) | 24.99 | 27.27 | |
| Ventilation time (days) | 2 (3.49) | 8.44 (6.35) | $p = 0.017$[*] |
| Time on ICU/IMC (days) | 7.92 (7.06) | 12.22 (3.13) | $p = 0.055$ |

**Notes.**
[#]Statistical analysis via Mann–Whitney-Test.
ns, non significant.
[*]$p \leq 0.05$.
[**]$p \leq 0.01$.
[***]$p \leq 0.001$.
[****]$p \leq 0.0001$.
miR-1246 expression presented as $2^{-dCT}$ values.

detected by musculoskeletal injuries, which are detectable for example *via* increased serum myoglobin level (*Ahmed et al., 2020*; *Giles et al., 2024*). The data of miR-1246 expression in this study showed a correlation with the systemic myoglobin. Some authors described a connection between miR-1246 and the differentiation of vascular smooth muscle cells (*Heo et al., 2020*; *Pan et al., 2021*), but so far, no connection between skeletal muscle and this miRNA has been investigated. Next to myoglobin, creatine kinase (CK) ($r = 0.43$) and CK-MB ($r = 0.49$) corelate with the upregulation of miR-1246 in polytraumatized patients of this analysis. In the clinical setting, CK and CK-MB could indicate not only injury severity due to skeletal muscle damage, but also due to cardiac contusion. Moreover, these biomarkers might help to detect a rhabdomyolysis or crush kidney. The literature has previously described a connection between the miR-1246 and heart injury: systemic plasma

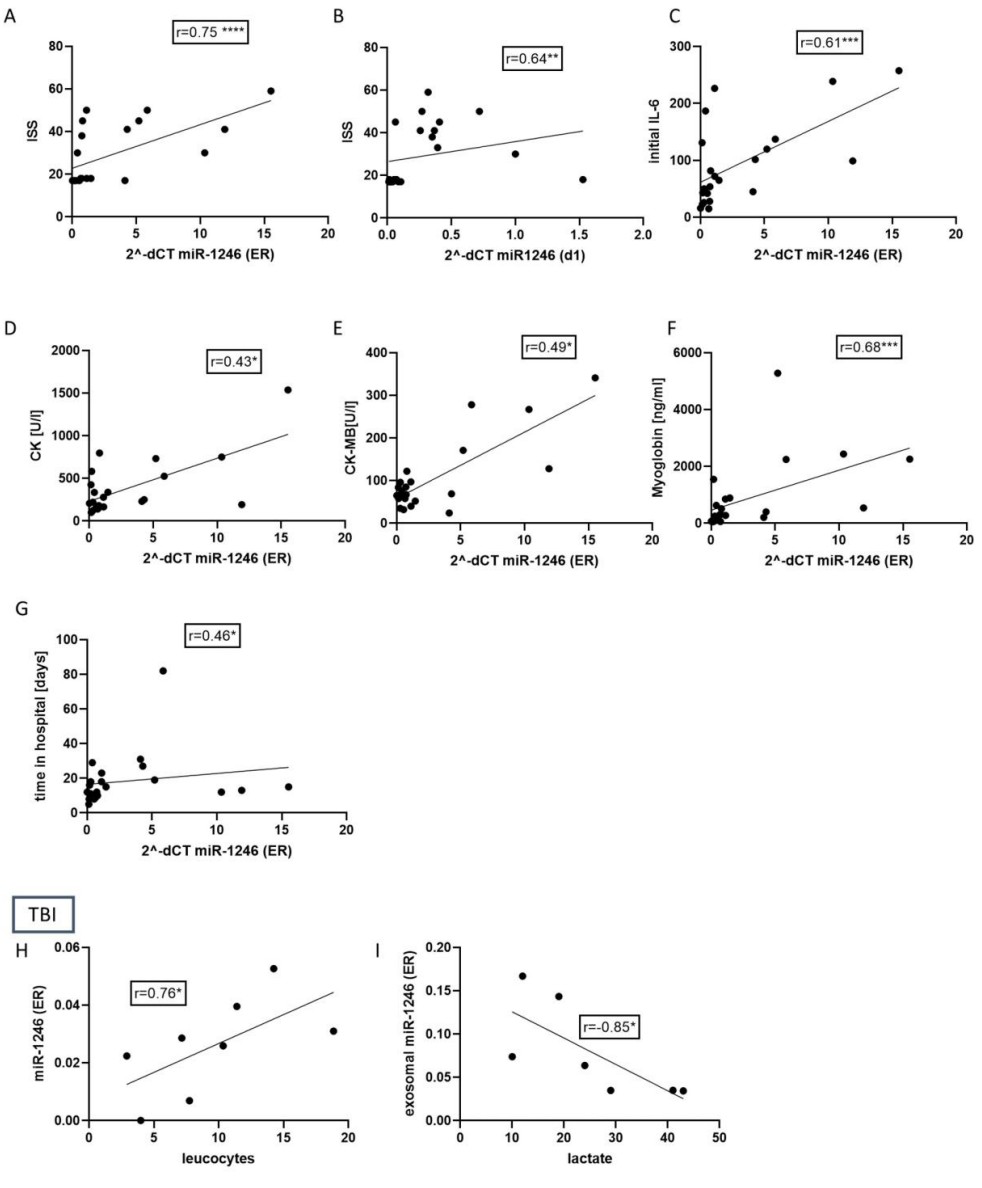

**Figure 3** **Correlation analysis of miR-1246 expression and patients' clinical data and routine laboratory parameters in polytraumatized patients and TBI patients.** (A) Strong correlation of miR-1246 and initial Injury Severity Score (ISS) ($r = 0.75$). (B) Analysis of correlation between miR-1246 expression measured 48 h after trauma and ISS ($r = 0.64$). Correlation miR-1246 with initial Interleukin (IL)-6 concentration (C), $r = 0.61$, creatinine kinase (CK) (D), $r = 0.43$, CK-MB (E), $r = 0.49$, myoglobin (F), $r = 0.66$, time in the hospital (G), $r = 0.46$. (H) Correlation analysis of miR-1246 expression in TBI patients and the number of leucocytes ($r = 0.76$) as well as the lactate concentration ($r = 0.85$). *$p < 0.05$; **$p < 0.01$; ***$p < 0.001$. Initial = Emergency room.

miR-1246 concentration was earlier implemented as biomarker of diastolic dysfunction in patients with cardiomyopathy (*Nair et al., 2013*). Nevertheless, in our present analysis, we did not observe any correlation between miR-1246 and troponin T, the gold standard of cardiac damage detection. This might discard its role as heart-damage biomarker in

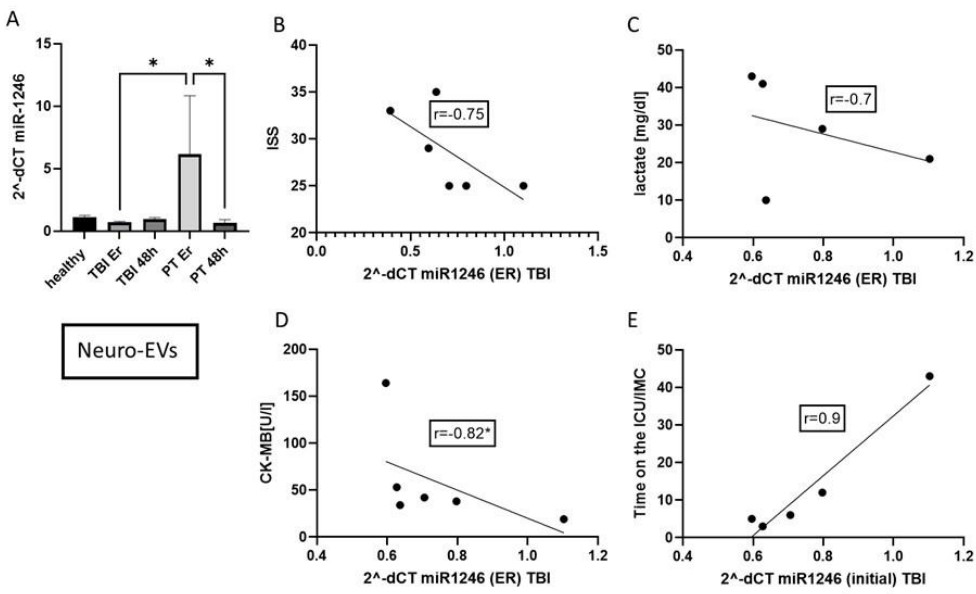

**Figure 4** **miR-1246 expression in neuro-EVs (CD171+).** (A) miR-1246 expression in neuro-EV isolated by magnetic-beads. (B) Correlation analysis of miR-1246 in neuro-EVs and Injury Severity Score (ISS). (C) Correlation of miR-1246 and lactate ($r = -0.7$), (D) creatine kinase muscle and brain subunit (CK-MB) ($r = -0.82$) and (E) time on the ICU/IMC ($r = 0.9$). *$p \leq 0.01$, initial = Emergency room.

for critically ill patients. Serum CK levels have been commonly used to screen patients with crush kidney injuries, to determine the injury severity and to predict resulting acute renal failure (reviewed in *Malinoski, Slater & Mullins, 2004*). So far, an association between miR-1246 and kidney function was only described in the EV compartment in the literature: miR-1246 was found to be significantly upregulated in patients with diabetic nephropathy and was related to albuminuria in these patients (*Kim et al., 2019*). In the urine extracellular vesicles of patients with autosomal dominant polycystic kidney disease, miR-1246 was also significantly up-regulated compared to healthy controls (*Ali et al., 2024*). In this study, we found a correlation of miR-1246 with ($r = .64$) the CK concentration in plasma ($r = 0.49$) 24 h after trauma, but not with the serum creatine, the standard marker of kidney damage. Whether miR-1246 could be a marker of rhabdomyolysis or crush kidney injury hence needs further investigations. Summarizing the above, ER miR-1246 plasma levels could function as potential biomarkers of trauma severity (ISS, CK or myoglobin) and inflammatory response (IL-6) in polytrauma patients.

### TBI patients

Interestingly, the systemic expression of miR-1246 in TBI patients was weaker compared to polytraumatized patients. But in general, the miR-1246 expression in TBI patients followed the tendency observed in polytraumatized patients (increase at ER). At the same time, the EV expression of miR-1246 in TBI patients significantly differed from the PT group. The EV expression of miR-1246 was significantly up-regulated not only at ER but also 48 h after trauma, which could reflect the secondary-injury and neuroinflammatory events of

TBI (*Kaur & Sharma, 2018*). EVs are complex signaling particles with the ability to directly transport biocompatible and active material across cells and initiate signaling events at the cell surface (*Colombo, Raposo & Théry, 2014*). They were shown to cross the blood brain barrier (BBB) in TBI patients and to promote the spread of neuroinflammation that may have systemic effects (*Kumar et al., 2017*; *Shao et al., 2022*). These facts offer a justification for their use as TBI biomarkers (*Schindler et al., 2024*). The enrichment of miR-1246 in glioma-derived exosomes has been observed in patients with glioma in the cerebrospinal fluid (CSF). This enrichment promotes the differentiation and activation of myeloid-derived suppressor cells, which help build the immunosuppressive milieu of glioma. In this study, the postoperative CSF exosomal miR-1246 expression was found to be associated with the glioma recurrence rate (*Qiu et al., 2021*). In this study's TBI patients, there was no correlation between the outcome of the patients (survival, time in the hospital, time on the ICU/IMC or ventilation time) and the miR-1246 concentration in the plasma-exosomes. CSF was not available in our study.

In addition, we observed a correlation between the number of leucocytes and plasma miR-1246 expression in TBI but not in PT patients. In the clinical setting, leucocytes have been used to detect inflammatory complications such as pneumonia, urogenital infections or sepsis. Following trauma, the typical response is an elevation of the total complete blood count (CBC) and a reduction of the lymphocyte count. This so called "leucocytosis" typically returns to normal values within 48 h after trauma and the persistence of a leucocytosis after polytrauma is associated with adverse outcomes and increased mortality (*Heffernan et al., 2012*). One of the explanations of the presence of a correlation between miR-1246 and leucocytes in TBI patients, and its absence in polytrauma patients, might be a different inflammatory kinetic in these two groups. In general, miR-1246 was associated with the inflammatory response in different studies. For instance, individuals with intracranial aneurysms had higher serum levels of miR-1246. Bioinformatics research showed that the target genes are primarily connected to lipid, atherosclerotic signaling pathways, and inflammatory response (*Jiang et al., 2022*).

Another notable observation in TBI patients is a strong negative correlation between the miR-1246 expression and the systemic lactate concentration. In the literature, patients with insufficient 24-hour lactate clearance showed a high rate of overall mortality, sepsis and non-septic complications (*Billeter et al., 2009*). In TBI patients, arterial lactate concentrations were associated with a poor systemic physiology, unfavorable clinical outcomes and disturbed cerebral blood flow autoregulation (*Svedung Wettervik et al., 2020*; *Younsi et al., 2023*). The noteworthy finding that miR-1246 in TBI patients correlated with trauma severity (by mean of correlation with ISS and IL-6) and negatively correlated with lactate could be a result of the specific treatment of these patients at ER aimed to prevent subsequent ischemic brain damage (reviewed in *Sontakke, Sontakke & Parihar, 2023*). An initial volume substitution (= infusion of crystalloid solution to restore blood pressure and blood volume) is the current standard of care for TBI patients and could be responsible for the altered lactate values in this group of patients. Lactate is now widely acknowledged as a key player in brain metabolism and several studies showed the importance of lactate and cerebral lactate metabolism following TBI (reviewed in *Patet et al., 2016*). Moreover,

neuroprotective effects of exogenous supplementation of lactate after TBI and stroke was found in several *in vivo* and clinical studies. All in all, the connection between TBI and lactate is discussed widely in the literature, but any connection with the miR-1246 or EVs was not described earlier.

## Neuro-EVs

The role of brain-derived EVs after brain injury is well established in recent research (reviewed in *Dong, Dong & Zhang, 2023*). In contrast, the connection between specific EV pool components with specific effects is still missing. The so called "brain-EVs" were shown to be able to cross the blood–brain-barrier and to induce cell apoptosis, endothelial damage and histological injuries in the lung, liver, heart, and kidney (*Li et al., 2023*), systemic coagulation (*Tian et al., 2015*). "Brain-EVs" negatively influence the brain itself by reducing cerebral blood flow (*Wang et al., 2022*). The literature describes a connection between brain-derived EVs and an early inflammatory response within the first 24 h after TBI was described (*Li et al., 2024*). Several studies tried to identify the roles of cell-specific EV fractions inside the brain EV pools. In one of our previous studies, we observed a significant increase of MOG+, MBP+, CD13+, CD133+ and CD196+ EV fractions in TBI patients (*Schindler et al., 2024*). This demonstrated the importance of oligodendrocyte- and neural stem cell -derived EVs in TBI.

As "brain-EVs" present a highly heterogenous pool of EVs originated from different types of brain cells, it is difficult to connect specific components to specific effects. This is first of all a result of an unknown characteristic signature of these EVs. Until now, the most common method for obtaining EVs of possible CNS neuronal origin has been the L1 cell adhesion molecule (L1CAM or CD171) (reviewed in *Gomes & Witwer, 2022*). However, recent findings showed that this marker is critical, as this protein is not restricted to neurons and can be present in biofluids as a cleaved, soluble protein (*Norman et al., 2021*). In a previous study, we were able to detect minor fractions of CD171 positive EVs in patients' plasma (*Schindler et al., 2024*) and therefore analyzed miR-1246 expression in the fraction of conventionally called neuro-EV or CD171+ (L1CAM) -EVs. We observed that in neuro-EVs, miR-1246 was only upregulated in the polytrauma-patients at the ER, reflecting the results seen for plasma miR-1246. Noteworthy, the results of a correlation analysis of neuro-EVs miR-1246 expression and clinical parameters differed from systemic or EV expression profile. miR-1246 in neuro-EVs negatively correlated with ISS ($r = -0.75$), lactate ($r = -0.7$) and CK-MB ($r = -0.82$). In addition, we found that neuro-EV miR-1246 expression strongly and positively correlated ($r = 0.9$) with the time of TBI patients on the ICU. These observations demonstrated that CD171+ EV could be a specific EV fraction, which significantly differ from other EVs by mean of miR-1246 cargo level and thus could play a unique role in the brain-EV pool. Future studies with access to improved cell-specific markers should focus on the identification of specific EV fractions inside the brain-EV pool and investigate their roles in intrinsic mechanisms of TBI pathophysiology.

## CONCLUSIONS

In this study, miR-1246 was shown to be a marker for a patient's injury severity (ISS, myoglobin), the early inflammatory phase (IL-6) and the patient's outcome (*e.g.*, length of hospital stay) after polytrauma. Furthermore, the EV miR-1246 has the potential to be used as a marker of secondary injury in TBI patients. Finally, we observed an increase of miR-1246 in CD171+ EVs of polytraumatized patients in the ER. These neuro-EVs might predict the time on ICU/IMC of polytraumatized patients and therefore need further attention.

### Funding

This work was conducted in the framework of the NTF consortium FOR5417/1 funded by the DFG (DFG, German Research Foundation)—project number 465409392. The funders had no role in study design, data collection and analysis, decision to publish, or preparation of the manuscript.

### Grant Disclosures

The following grant information was disclosed by the authors:
DFG (DFG, German Research Foundation): 465409392.

### Competing Interests

Liudmila Leppik is an Academic Editor for PeerJ.

### Author Contributions

- Liudmila P. Leppik conceived and designed the experiments, analyzed the data, prepared figures and/or tables, authored or reviewed drafts of the article, and approved the final draft.
- Melissa Manamayil performed the experiments, analyzed the data, prepared figures and/or tables, and approved the final draft.
- Cora Schindler performed the experiments, authored or reviewed drafts of the article, and approved the final draft.
- Ramona Sturm performed the experiments, authored or reviewed drafts of the article, and approved the final draft.
- Philipp Störmann performed the experiments, authored or reviewed drafts of the article, and approved the final draft.
- Dirk Henrich conceived and designed the experiments, authored or reviewed drafts of the article, and approved the final draft.
- Ingo Marzi conceived and designed the experiments, authored or reviewed drafts of the article, and approved the final draft.
- Birte Weber conceived and designed the experiments, analyzed the data, prepared figures and/or tables, and approved the final draft.

## Human Ethics

The following information was supplied relating to ethical approvals (i.e., approving body and any reference numbers):

The study was conducted in accordance with the Declaration of Helsinki, and approval was given by the Local Ethics Committee of the University (approval ID 89/19 and 375/14).

## Data Availability

The raw data is available in the Supplemental Files.

## Supplemental Information

Supplemental information for this article can be found online at http://dx.doi.org/10.7717/peerj.19185#supplemental-information.

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
