# Peer review of "New role for an old acquaintance: miR-1246 as a new inflammatory and prognostic marker in polytrauma patients"

_PeerJ, doi:10.7717/peerj.19185_

## Round 0.1 · original submission · Major Revisions

The manuscript has been reviewed by two reviewers, both of whom have requested major revisions. Their comments and suggestions can be found below. Please carefully review the feedback and provide detailed responses to their critiques while revising the manuscript accordingly.

Reviewer 1 ·

Basic reporting

In this paper, the authors revisited the use of a miRNA (miRNA-1246), previously described in the context of cancer, and investigated its potential implication and correlation with inflammation in the context of polytrauma and traumatic brain injury. The authors provided a brief yet comprehensive overview of previous findings with miRNA-1246 and have successfully gathered data from healthy donors, patients with polytrauma and TBI, and compared miRNA-1246 levels in plasma, EV and CD171+neuro-EVs. The experimental design to investigate the correlation between trauma and miRNA-1246 is sound, however I have several comments:

In the text, all abbreviations should be introduced as full words when first used, for example TBI, ALI, ISS, AIShead).

I recommend that the authors be consistent with the writing of miRNA-1246 and miR-1246 across the manuscript. In the introduction, miRNA-1246 is used but the authors switch to miR-1246 later in the manuscript.

Line 157, should be “and” not “und”

The writing is clear and conveys the authors’ claims

Experimental design

The authors mentioned that plasma was collected at admission to the ER, as well as at 24 and 48 hours. However, the results at 24 hours are not presented in Figure 1 and could be valuable too. This data could be presented either in the same figures or as a supplementary.

In Figure 1, authors report values at admission and 48 hours while in Figure 2 values are reported at admission and 24 hours. I suggest the authors show the same timepoints consistently across the article.

The authors suggest that miRNA-1246 correlates with trauma severity. Figure 2 alludes to this, but to support the claim, a comparison of miRNA-1246 levels should be made between patient with ISS<20 and ISS>20 at each timepoint. The authors uniquely show the comparison to healthy controls, which illustrates an increase in miRNA-1246 levels with trauma but no specific correlation with injury severity.

The results presented are variable between polytrauma patients and patients with TBI, which makes it difficult to interpret. The authors suggest that the difference might be due to differences in trauma severity. Although there doesn’t seem to be a difference in injury severity based on table 1, the reason behind splitting the cohort into patients with and without TBI is unclear. The hypothesis the author claimed was that miRNA-1246 correlated with injury. It might be worthwhile to combine polytrauma and TBI patients into a single cohort and splitting based on median ISS for example.

Validity of the findings

Based on the introduction, it seems that results with miRNA-1246 have been inconsistent across studies. The authors mention “in one patients study, miR-1246 was found to be increased in patients with community-acquired pneumonia and pneumonia-related sepsis”, while “another study with patients showed that miR-1246 is downregulated in sepsis and sepsis-acute lung injury”. Taking these inconsistencies into consideration, it would be important for authors to clarify why they still believed that investigating this miRNA specifically was important, why not another one.

It is suggested in the manuscript that patient with polytrauma have increased plasma miRNA-1246 at ER admission. However, the cohort of patients with TBI also had polytrauma, why didn’t they also have upregulation of miRNA-1246 at admission? There seem to be a big difference between TBI patients and those with polytrauma uniquely.

Authors would benefit of mentioning the p values in the results section when discussing comparisons or illutrasting correlations.

While miRNA-1246 in plasma correlated with ISS and CK-MB, the opposite trend is observed in CD171+Neuro-EVs. Is there any explanation or hypothesis for this?

In the discussion, the authors mention “level of miR-1246 was increased in both groups of patients [...] at ER.” (lines 209-210). Based on Figure 1, this doesn’t seem to be the case for patients with TBI. There’s only slight elevation in miRNA levels, and statistics are not shown.

In the discussion, the authors mention that the higher levels of miRNA-1246 in polytrauma patients is likely due to the more severe injury in this cohort (lines 212-213). However, Table 1 shows that the ISS score between the two cohorts is similar (p value non-significant).

The authors mention that parameters such as ISS and IL-6 have been demonstrated to strongly predict patient outcomes in trauma. In this article, the authors show a correlation between miRNA-1246 and these parameters at admission. Yet, miRNA-1246 seems to exhibit more variability and lacks standardized measurement methods. At this stage, the clinical usefulness of miRNA-1246 compared to those other clinical parameters remains unclear. The authors could explore how well miRNA-1246 levels correlate with outcomes (complications, length of hospital stay, etc.), compared to those other established parameters (ISS, IL-6 etc.).

The authors mentioned that miRNA-1246 levels in TBI patients “correlated iwth trauma severity” (line 254). This was not shown, relation to injury severity with ISS was discussed for patients with polytrauma, not for patients with TBI. The authors would benefit from showing this correlation. In addition, there doesn’t seem to be a significant difference in miRNA-1246 in plasma or EV of patient with TBI compared to healthy donors at admission.

It is surprising to see that neuro-EVs are uniquely upregulated in polytrauma patients and not in TBI patients. One would expect that those coming from the brain would be affected by brain injury.

Reviewer 2 ·

Basic reporting

The study presented by the authors investigates the potential of miR-1246 as a biomarker for the severity of traumatic brain injury (TBI) and polytrauma in patients. The hypothesis that a microRNA could reflect the intensity of such injuries holds significant clinical implications, as it may provide a rapid and non-invasive indicator for monitoring patient progression. The manuscript is written in professional English, and the language is clear and understandable. However, a typographical error on line 51 should be addressed. Furthermore, the authors have made the raw data available, which is a positive aspect in terms of transparency and reproducibility of the study.

Experimental design

The study presents several important findings, but there are notable gaps and missing data that need to be addressed to strengthen its conclusions. Firstly, the characterization of exosomal markers, particularly CD171, was referenced but not substantiated with results, such as Western Blot analysis, which is essential for confirming the presence of these markers. Additionally, TEM or Nanoparticle Tracking Analysis are crucial methods for the morphological and quantitative characterization of exosomes, yet no data were provided in this regard. Furthermore, the role of NF-kB signaling and TNF-α expression in the context of injury severity remains underexplored, particularly in relation to levels of these markers in both exosomes and plasma. It is unclear how the injury severity score was calculated, and further clarification of this metric is necessary for reproducibility and interpretation. Additionally, discrepancies in the qPCR data are evident, as some results differ by more than one cycle in the Cq value, which could indicate issues with assay sensitivity or sample quality. The study also faces limitations in the amount of neuro EV samples, which are reported to be at low quantities, leading to weak and often non-significant correlations driven by a single data point. Increasing the sample size would likely enhance the statistical power and reliability of the findings, providing more robust and reproducible results. These points need to be addressed in future iterations of the study to ensure the validity and generalizability of the conclusions.

Validity of the findings

The need for reliable biomarkers in the diagnosis and management of trauma is undeniable and represents an important area of research. The study under review addresses this crucial topic, proposing miR-1246 as a potential biomarker for traumatic brain injury and polytrauma. However, despite the intriguing premise, the study requires substantial technical improvements to enhance the robustness and reliability of the data. Key methodological issues, such as the lack of comprehensive exosomal marker characterization (e.g., CD171) via Western Blot, as well as the absence of crucial imaging and quantification techniques like TEM and NTA, undermine the strength of the findings. Additionally, discrepancies in the qPCR data, the low quantity of neuro EV samples, and the limited statistical significance of the correlations further diminish the overall validity of the study. Addressing these technical limitations, including increasing the sample size and improving assay consistency, would substantially strengthen the study’s conclusions and its potential impact in the field.

Additional comments

Sample Size and Statistical Power: As previously noted, the sample size, particularly for neuro EV samples, is relatively small, which limits the statistical power of the study. Increasing the number of samples would help reduce variability and improve the reliability of the results. Additionally, more robust statistical analysis, such as correction for multiple comparisons, would strengthen the conclusions and reduce the likelihood of findings being driven by outliers or random fluctuations.

Standardization of Methodology: The study would benefit from a more detailed description of the methodologies used, particularly regarding the calculation of the injury severity score. Standardizing this metric across different studies would improve the reproducibility of results and provide a clearer understanding of the relationship between biomarkers and trauma severity.

Technical Validation of Exosomal Markers: The absence of comprehensive validation for exosomal markers, such as CD9, CD81, CD63, CD171, via techniques like Western Blot, and the lack of imaging and quantification using TEM and NTA, limit the strength of the findings. Incorporating additional methods such as flow cytometry or mass spectrometry would provide further confidence in the identification and integrity of the exosomes analyzed.

Mechanistic Insights: While miR-1246 is the focus of this study, the underlying biological mechanisms driving its potential as a biomarker remain insufficiently explored. A more in-depth analysis of how miR-1246 may be involved in the pathophysiology of traumatic brain injury and polytrauma—particularly in relation to inflammatory pathways or cellular signaling (e.g., NF-kB, TNF-α)—would offer greater insight into its utility as a biomarker.

---

## Round 0.2 · Minor Revisions

Dear authors,

I congratulate you on your dedication and patience. I highly value the reviewer’s minor revision suggestions. I hope to share positive news with you after these suggestions are clarified.

Reviewer 1 ·

Basic reporting

Please clarify in Figure 3 whether the upper panel relates to polytrauma patients only. It is mentioned that Figures 3H and I are for TBI patients, the authors would benefit from clarifying on the Figure whether the upper panels (A-G) relate to PT or PT+TBI.

Experimental design

NA

Validity of the findings

The authors mention: "miR-1246 was expressed significantly higher in patients with more severe trauma at both time points (Figure 2A) compared to healthy controls and patients with ISS<20." This is inaccurate: First, comparison on Figure 2A is only made to healthy controls, there are no comparisons between ISS<20 and ISS>20. Second, while miR-1246 seems higher in patients with ISS>20 compared to ISS<20 on ER, this does not seem to be true on day 1.

Reviewer 2 ·

Basic reporting

all requests are respect

Experimental design

The authors improved the experimental design. In my opinion, the criteria for publication are fulfilled

Validity of the findings

no comments

---

## Round 0.3 · accepted · Accept

Congratulations on your successful research.